# Lights and Shadows in Immuno-Oncology Drug Development

**DOI:** 10.3390/cancers13040691

**Published:** 2021-02-09

**Authors:** Milana Bergamino Sirvén, Sonia Pernas, Maggie C. U. Cheang

**Affiliations:** 1Clinical Studies and Clinical Trials and Statistics Unit, The Institute of Cancer Research, London SM2 5NG, UK; 2Department of Medical Oncology, Catalan Institute of Oncology—ICO, L’Hospitalet de Llobregat, 08908 Barcelona, Spain; spernas@iconcologia.net; 3Breast Cancer Group, Institut d’Investigacio Biomedica de Bellvitge—IDIBELL, L’Hospitalet de Llobregat, 08908 Barcelona, Spain

**Keywords:** immuno-oncology, cancer, trial design, endpoints, biomarkers

## Abstract

**Simple Summary:**

The introduction of immunotherapy has had a significant impact on the cancer treatment landscape, with unprecedented survival outcomes in some tumor types. However, clinical development of immune-oncology (IO) agents presents both opportunities and challenges, and not all patients benefit to the same extent. Many factors influence trial designs and could potentially threaten the success of promising IO drugs: 1. Most IO trials still rely on response evaluation criteria based on image assessment only, while new approaches including biomarkers tracking response should be incorporated. 2. Surrogate endpoints for efficacy are still inferred from classical anticancer drugs that have not been specifically validated for IO trials. 3. There is a need for biomarker-driven clinical studies in order to select appropriated patients. 4. Long-term toxicity monitoring is needed, and dosage calculation should not rely on dose-dependent toxicities. 5. Optimizing the design of new IO agents with collaborative approaches assessing multiple drugs on a biomarker-based basis is needed.

**Abstract:**

The rapidly evolving landscape of immuno-oncology (IO) is redefining the treatment of a number of cancer types. IO treatments are becoming increasingly complex, with different types of drugs emerging beyond checkpoint inhibitors. However, many of the new drugs either do not progress from phase I-II clinical trials or even fail in late-phase trials. We have identified at least five areas in the development of promising IO treatments that should be redefined for more efficient designs and accelerated approvals. Here we review those critical aspects of IO drug development that could be optimized for more successful outcome rates in all cancer types. It is important to focus our efforts on the mechanisms of action, types of response and adverse events of these novel agents. The use of appropriate clinical trial designs with robust biomarkers of response and surrogate endpoints will undoubtedly facilitate the development and subsequent approval of these drugs. Further research is also needed to establish biomarker-driven strategies to select which patients may benefit from immunotherapy and identify potential mechanisms of resistance.

## 1. Introduction

Immuno-oncology (IO) is redefining the cancer treatment landscape and the way that some solid tumors are treated. Almost 5000 new agents from six different main classes of immunotherapies have been in the drug development pipeline within 2020, including adoptive cell therapy, cancer vaccines, T cell-targeted immunomodulators, other immunomodulators, oncolytic viruses and antibody-based targeted therapies [1]. Overall, IO consists of a wide range of drugs with different mechanisms of action that ultimately lead to the enhancement of immunity against tumor cells. The immune checkpoint inhibitors, which reactivate T-lymphocyte mediated immune response against tumor cells, have been the most successful type of IO developed since the beginning of the “IO era”. Their use is now a standard of care across several solid tumors, including melanoma, non-small cell lung cancer (NSCLC), gastric cancer, head and neck squamous cell carcinoma, renal cancer, bladder cancer, cervical cancer and triple-negative breast cancer (BC) among others [2,3,4,5,6].

### 1.1. Which Are the Current Challenges with IO?

A substantial number of IO therapies do not progress from phase I-II trials, and some fail in late-trial stages [7]. First, clinical trials testing novel IO drugs still rely on biomarkers and endpoints that are validated for conventional treatments such as cytotoxic agents, although their mechanisms of action are different [8]. Second, the adaptative immune response induced by immunotherapeutic agents is often sustained in time, as well as the inflammatory and autoimmune-related adverse events. Those long-term effects make the evaluation of clinical benefits and toxicities extremely difficult [9]. A wide range of exclusive toxicities to IO agents, mainly characterized by significant latency, has been underestimated by many clinical trials as they are only evident in long-term follow-up or pharmacovigilance studies [10,11,12]. In addition, the calculation of the IO drug doses still relies on dose-limiting toxicities seen in the first cycles of classical anticancer treatments, which seems inadequate for IO treatment. Furthermore, the differences in the intrinsic biology between IO agents and conventional therapies further complicate their comparisons. Finally, some particular pathways in IO are overcrowded with similar drugs within the same therapeutical setting but led by different pharmaceutical companies. However, these resources could be better relocated to the development of new biomarkers to select the best in class and to test mechanistically different drugs.

BC, for example, is much in need of a paradigm shift of better IO drug development [13]. This cancer type belongs to a group of widely considered “cold tumors”, characterized by low mutation and neoantigen burden and low counts of tumor-infiltrating cells (TILs). Although the number of clinical trials assessing the use of immunotherapy in BC is increasing, to date, the approval of its use is only for a selected subset of advanced triple-negative (TN) BC patients with >1% of programmed death ligand 1 (PD-L1) expression by immunohistochemistry (IHC) [6,14]. Overall, the main problem in the development of IO agents in this type of cancer has been the lack of biomarker-guided patient selection for trials and the reliance on a reduced number of “classic” biomarkers such as PD-L1. In particular, PD-L1 remains at least insufficient to fully explain the therapeutic success and durable clinical benefit seen in some patients with PD-L1 non-expressing tumors, especially when treated with other IO agents beyond checkpoint inhibitors [15,16,17,18,19]. However, new genomic alterations such as those in DNA damage response or specific mutated gene pathways have shown promising results as immunomarkers in some translational studies, and their validation in clinical trials should be encouraged [20]. In addition, the main focus of IO development in BC has been put in TNBC due to its general enrichment of TILs and the lack of effective therapies other than chemotherapy. Although the use of IO agents in other subsets of BC, such as luminal B tumors or pretreated HER2-positive BC, has also been explored, there is still much controversy on its use in those settings [21,22,23]. It is still unclear whether a better design based on a more accurate selection and less pretreated patients would have led to positive results. The other main issue is the lack of assessment of IO agents’ combinations, which is now believed to be an alternative strategy to achieve immune response enhancement in “colder tumors”. Especially in BC, in which many different pathways, such as estrogen receptor signaling, seem to have major implications for the tumor immune scape, and further combinations of different IO treatments with classical anticancer therapies could potentially help to overcome them [24,25].

### 1.2. How Can We Do Better?

Despite the great improvement in the field of IO in the past years, most new agents still offer a modest rate of objective responses and poor long-term outcomes compared to conventional treatments. In addition, immune-mediated serious adverse events remain a potential issue [26]. In order to optimize new IO trials’ design to further improve survival outcomes and minimize toxicity rates, new strategies are needed. We have identified five main domains in the development of promising IO treatments that should be redefined for more efficient results and accelerated approvals.

First, there is a crucial need for a biomarker-driven selection of a patient population for each of the new IO agents. Second, the evaluation of efficacy, toxicity and comparisons with current treatments should be based on surrogate endpoints and response criteria for those specific IO agents and not just inferred from conventional drugs. Monitoring long term-outcome seems to be mandatory as IO agent efficacy and toxicity may have long-term effects [27]. Finally, changing the current scenario of IO clinical trial-design with more collaborative approaches that facilitate the assessment of multiple diseases and drugs on a biomarker basis seems crucial [28].

In this manuscript, we will analyze each of those points and suggest some potential improvements in the field.

## 2. Challenges and Opportunities in IO Drugs Development

A summary of the main critical aspects in IO drug development and strategies for their optimization is illustrated in Figure 1 and Figure 2, respectively.

### 2.1. Response Criteria

From the beginning of the “IO era,” researchers realized that the standard response evaluation criteria in solid tumors, namely RECIST alone, would not be suitable for immunotherapy due to the indirect effect caused by the participation of inflammatory cells and their interactions [29]. RECIST relies on the early suppression of tumor growth by chemotherapy and may consequently underestimate the benefit of immunotherapy. The immune RECIST (iRECIST) and other immune-specific related response criteria were developed to evaluate the heterogeneity of responsiveness in patients receiving immunotherapy [30]. Although iRECIST takes into account pseudoprogressions or hyperprogressions, which are observed exclusively with IO agents, demonstration of actual response to treatment may not be distinguishable from such patterns for another several months. This particular timeline may have caused many clinicians to have considered patients as no treatment response and/or stable when the patient may have conversely obtained benefit from treatment since iRECIST still recommends that assessment of response durability may occur between 4 and 8 weeks [31].

The use of iRECIST for evaluation of response has mainly been used for studies evaluating the efficacy of immunotherapeutic agents only. It becomes rather complex when the primary objectives of trials are to perform head-to-head comparisons with non-immunotherapeutic agents or combinations with other cytotoxic or molecular targeted treatments. On the other hand, particular responses observed in patients undergoing immunotherapy treatments are not well captured by iRECIST, such as dissociated responses with some lesions growing, some shrinking or the slow progressions, features linked with clinical benefit. In those cases, classic RECIST could still remain as a meaningful method of evaluation [32]. In addition, most IO clinical trials that compare immunotherapy with other cytotoxic antitumor agents are designed around the evaluation of response on superiority, inferiority or equivalence [33]. These comparisons may not be appropriate when assessing two mechanistically different treatments, in which case integrative clinical benefit and long-term outcome should be taken into consideration. Although the American Society of Clinical Oncology-Society for Immunotherapy of Cancer (ASCO-SITC) has recently recommended reporting responses according to both the conventional RECIST and iRECIST criteria in parallel [34], this approach is still suboptimal as their evaluation is done independently. Harmonizing and integrating both measurement methods into an adequate tool rather than just separately considering both criteria is urgently warranted for more precise measurements of actual responses.

It is worth noting that some methods beyond image evaluation have also shown promising results for the assessment of response in IO. Some biomarkers assessed in peripheral blood such as interleukin 8 (IL-8), tumor circulating DNA (cDNA) or CD8+ memory effector cytotoxic T cells have recently shown to assist in tracking immune response in different tumors like NSCLC cancer and melanoma [35,36,37]. Given the strong correlation between detected changes on those biomarkers with subsequent clinical responses, biological assays exploring changes in their expression levels could lead to promising results. Based on that, decisions could be made on the basis of patient-specific immunological tracking biomarkers. Moreover, the pathological assessment of lymphocytes and immune infiltrating cells in tumor biopsies during treatment could also be considered as a useful monitoring tool of clinical response in clinical trials and help to gain a deeper understanding of the changes in tumor characteristics under immunotherapy. However, the application of on-treatment biopsies in clinical practice remains unclear and should be further investigated and refined as it could be considered invasive [38].

In summary, combining an integrative image tool with response markers such as liquid biopsies could become better tailoring of response evaluation in the future of IO.

### 2.2. Long-Term Efficacy Endpoints and Surrogates

Drug approval is generally based on safety and efficacy assessed by clinically relevant endpoints in phase 3 randomized trials. Overall, survival (OS) is considered to be the gold standard as it reflects the ultimate survival benefit from cytotoxic and other targeted therapy regimens, and there are minimal measurement errors in OS. Meanwhile, progression-free survival (PFS) and objective response rate (ORR) are used as surrogate endpoints in cancer; these measurements provide inferred conclusions from clinical trials and facilitate accelerated approval of new drugs that fill an unmet clinical need [39,40]. Other emerging biological endpoint, such as changes in Ki67 level after short-term treatment with endocrine therapy in BC, has been validated as a surrogate of long-term benefit [41] and are increasingly becoming used as primary endpoints in clinical trials. Surrogate endpoints for the assessment of IO agents’ efficacy have primarily been adopted for cytotoxic and molecular targeted drugs; questions remain whether they are suitable to determine benefit from immunotherapy. In particular, there are increased doubts concerning the use of short-term benefit endpoints such as PFS or ORR as primary endpoints in most clinical trials of new IO agents [7,42,43].

A recent meta-analysis of 60 published immunotherapy randomized clinical trials suggested that ORR could be a meager surrogate of response to evaluate the efficacy of IO, and the use of PFS as a surrogate of OS is still indeterminate [44]. Another meta-analysis of 12 randomized clinical trials did not find a significant positive correlation between the OS and PFS hazard estimates, suggesting that PFS assessment is not sufficient to capture the benefit of PD-1-inhibitors in patients with solid tumors [42]. This is not surprising as the unique mechanism of immunotherapy’s impact on tumors shows different patterns of response and progression from other conventional agents [45]. Emerging biological endpoints such as changes in Ki67 level are mainly driven by the antiproliferative effect of certain drugs; however, its use in trials evaluating the combination of IO agents with cytotoxic and targeted drugs should be further explored. Recent studies have also shown that endpoints taking into account a component of the duration of response such as milestone survival or durable response rate may better capture the delayed and persistent responses derived from IO agents and should be further studied [46]. For instance, milestone survival is the survival probability at a given time point, defined a priori as two years, and durable response rate can be measured as a continuous response, such as complete or partial objective responses, beginning within 12 months of treatment and lasting ≥6 months [47]. The advantages of such types of integrative endpoints are that they take into account particular behaviors seen exclusively in IO and allow the use of predefined cutoff time intervals that lead to the rapid characterization of survival probability and inference to long-term survival data.

To date, overall survival still remains the gold standard for the evaluation of clinical efficacy of IO agents in late-phase clinical trials, new biomarker-driven surrogate endpoints that capture the mechanisms of action in early phases of immunotherapy drugs development should be explored.

### 2.3. Biomarkers of Response

Despite the rapid advance and reduced cost of high-throughput sequencing technologies, most current trials in oncology have limited use of biomarker-based selection and stratification strategies in their designs. However, studies in IO treatments tested in unselected populations are generally negative [48,49]. The lack of new and robust predictive markers is particularly concerning for the selection of the most appropriate subpopulations in relative “colder tumors” such as BC, in which most patients will not benefit from those new IO agents.

In recent years more attention has been paid to the identification of predictive biomarkers of the efficacy of IO drugs to identify patients who benefit from those agents. Most approved immunotherapeutic treatments show efficacy only in selected populations, mainly based on the immunohistochemical (IHC) levels of PD-L1 checkpoint target leading to PD-L1 being the compulsory companion diagnostic assay for the administration of many checkpoint inhibitors in oncology. Many efforts are currently focusing on the reproducibility and standardization of laboratory protocols for the IHC assessment. However, the evaluation of single IHC biomarkers does not completely explain the heterogeneity of tumors, and their use seems to be suboptimal in the “genetics and omics era”. In particular, multiplex diagnostic assays would be better, especially when testing IO agents beyond immune checkpoint inhibitors [14,15,16,17,18,19].

Emerging predictive biomarkers as defined by both the host and tumor factors are promising measurement for a clinical response; the use of these measurements are still in infancy stage due to a lack of standardization and harmonization of reporting methods [50]. Tumor mutational burden, microsatellite instability, and tumor neoantigen loads are some examples. Mutational burden and high microsatellite instability assessment based on mutations in mismatch repair genes have been associated with better response to immunotherapy, especially to anti-programmed cell death protein 1 (anti-PD1) agents [51,52]. Next-generation sequencing has led to a more accurate method of quantification, but it is still difficult to achieve a homogenization on their quantification. Thus, the implementation and standardization of robust bioinformatics methodologies and analytical techniques across laboratories are necessary [53]. Additional exploration to identify the type of mutations is much needed for generating the most relevant neoantigen for recognition by the T cells. Other challenges are that these emerging predictive biomarkers assays for IO agents are usually expensive, technically demanding and not widely available.

Multi-plex and multi-omics based biomarkers indicating higher tumor immune tolerance such as immune-related genes and signatures have thrown some light on the field of predictive biomarkers in IO [20,54]. Some studies have shown that high expression of some particular gene expression signatures is associated with response to PD-L1 inhibitors regardless of their PD-L1 status in NSCLC and melanoma [55]. Other signatures, including some targetable immune checkpoint components such as indoleamine 2,3-dioxygenase (IDO1), lymphocyte-activation gene 3 (LAG-3), or interferon-gamma (IFNγ) genes can predict benefit from immunotherapy in “colder tumors” such as luminal B BC patients [56,57]. Recent studies have also demonstrated that exhausted CD8+. T cell signatures can predict immunotherapy response in ER-positive BC [58]. These signatures are yet to be validated in clinical trials.

Identification of robust pharmacodynamic biomarkers of IO response remains a challenge [59], and it is likely that combinations of two or more biomarkers to capture immune status more accurately will be needed [20].

### 2.4. Definition of Toxicity and Treatment Dosage

Immunotherapies have the potential to induce toxicity profiles distinct from those from other cancer treatments. Immune-related adverse events are often underdiagnosed, as patients can remain asymptomatic for long periods of time [10,12]. In addition, toxicity is mostly inflammatory-related, and assumptions from cytotoxic or molecular targeted treatments are not appropriate for IO agents. In particular, precise considerations must be taken during their development [60] as they do not fit the dose–response/dose-toxicity relationships seen with cytotoxic therapies.

In contrast to what usually happens with cytotoxic drugs, an increase in the dosage above the biologically optimal does not always correlate with an increase in efficacy or toxicity in IO [7]. Due to the lack of reliable toxicity endpoints to establish optimal dosages in immunotherapeutic trials, some FDA approvals on immune-checkpoints inhibitors have been based on varying dosages and schedules across different tumor types. This has resulted in some confusion on the clinical implication and implementation for future trial designs [32]. The best approach to define the optimal administration dosages and schedules with the highest efficacy is still open-ended, and deficient toxicity profiles may have unfortunately an impinged on the results from several clinical trials assessing efficacy with IO agents [61].

The optimal design of early phase clinical trials should aim to evaluate doses and schedules at the minimal doses that are biologically active. Based on the distinct behavior of IO agents, flat dosing administration instead of weight-based dosing may be a better approach facilitating smoother administration and avoiding drug waste [62,63]. Long-term follow-up of IO related adverse events is encouraged in trial designs evaluating new-IO agents. Consensus guidelines for recognizing each of the adverse events under immunotherapy and specific management of these reported events should also be incorporated.

Aforementioned, defining doses and schemes of IO agents seem relatively more complicated than with cytotoxic drugs. New approaches in trial designs, including the homogenization of the optimal dosages that will be carried over later phases of drug development, are urgently needed. Finally, the incorporation of additional endpoints especially validated for IO agents in early phases of trials for dose selection to improve efficacy and reduced toxicity, are also warranted.

### 2.5. The Trial Design Itself

Due to the particular impact on tumor biology by IO treatments, conventional phase 3 clinical trial designs to demonstrate the effects of an experimental therapy compared to standard of care are unlikely to provide definitive answers on the efficacy of IO within reasonable time and cost. The anti IDO1 epacadostat, which was evaluated in a late-phase trial in melanoma, is an example of such a conundrum [6]. This phase III trial ECHO-301/KEYNOTE 252 trial was designed to assess the efficacy of IDO1 inhibition in combination with pembrolizumab, but there were several problems associated with the trial, including the use of endpoints such as PFS and ORR and no pre-planned translational studies to study the tumor biology leading no collection of biological samples that could be studied further to explain the unexpected clinical results [62,63]. This study has posed that translational studies are important elements to be incorporated in the trial design whenever possible.

Furthermore, the field of IO is currently overcrowded with several drugs competing for the same therapeutic space. Advancements in “precision oncology” urge therapy selection based on tumor molecular characteristics. The conventional trial designs, lack of pairing tumor characteristics with therapeutic targets, are not adequate to investigate the broad-spectrum of genetic makeup in tumors that may benefit from different IO agents and targeted therapies. The incorporation of “master protocols” in collaborative clinical study designs can allow multiple disease assessments and multiple strategies at a time. Some “modern” strategies also include several trial designs that enable more personalized and adaptative assessment of new drugs, such as platform trials, which can be multi-arm, multi-stage adaptive studies, pairing targeted therapy with molecular characterization of tumors [64]. Other simpler approaches include umbrella trials, which evaluate multiple targeted therapies for a single disease as defined by specific molecular characteristic subgroups, and basket trials, which use biomarkers for molecular screening and allocation of patients into different trials according to their molecular biology [65,66].

These new approaches in trial designs alleviate the field of IO by optimizing resources and improving the efficiency of the broad amount of emergent clinical trials. However, those new designs require considerable effort, cost and multidisciplinary collaboration. Regulatory approvals and standardization of their use warrant further exploration. The question remains whether a systematic implementation of renewed IO drug designs and translational studies should be encouraged to avoid superfluous numbers of clinical trials.

## 3. Conclusions

The current clinical development of IO agents has both strengths and weaknesses that provide us with challenges and opportunities for improvement. Given the rapid growth in the IO field, only a greater level of understanding of the underlying mechanisms of resistance, tumor heterogeneity and host and tumor microenvironment can shed light on IO for patient selection through robust biomarkers testing. The silhouette in clinical trial design and response evaluation criteria of IO should be redefined with new approaches like amalgamating integrative tools with response biomarkers such as the use of liquid biopsies. Applying innovative but appropriate clinical trial designs, incorporating multiple robust biomarkers assays of response and surrogate endpoints, will lead to the best possible development and accelerated approvals of new IO agents that are here to stay. Finally, the precise definition of adverse events following long-term evaluation and dosage definition is also needed for successful results. With advances in molecular sequencing technologies and development in machine-learning methods, biomarker-driven strategies to assist the selection of patients for future trials with immunotherapy will soon be a reality.

## Figures and Tables

**Figure 1 cancers-13-00691-f001:**
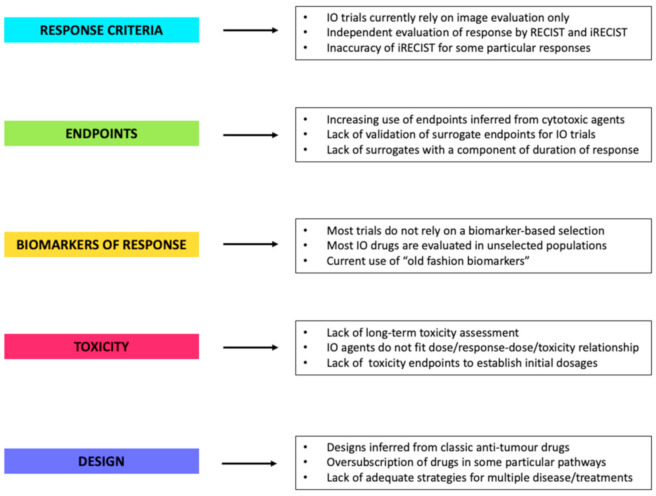
The major drawbacks found in immune-oncology trials designs to date. Abbreviations: IO: immuno-oncology, RECIST: response evaluation criteria in solid tumors, iRECIST: immune response evaluation criteria in solid tumors.

**Figure 2 cancers-13-00691-f002:**
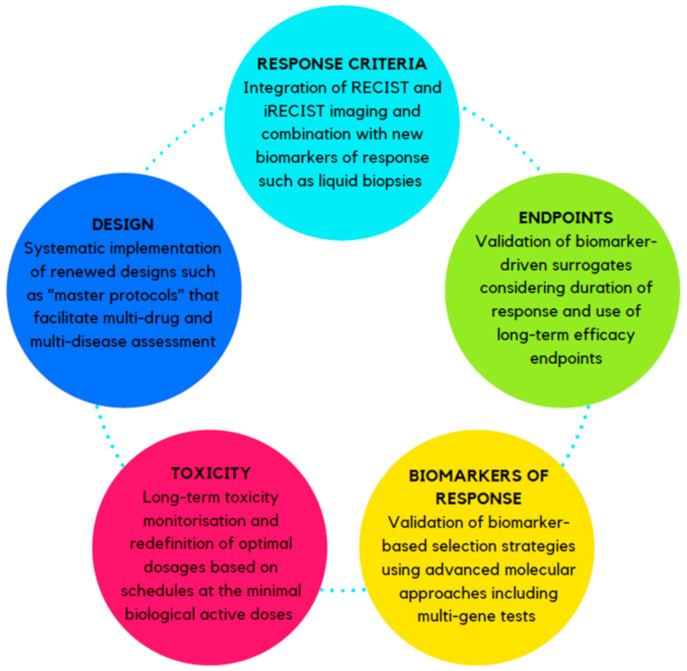
Proposed strategies to optimize the clinical trial design for new immuno-oncology agents. Abbreviations: RECIST: response evaluation criteria in solid tumors, iRECIST: immune response evaluation criteria in solid tumors.

## Data Availability

No new data were created or analyzed in this study. Data sharing is not applicable to this article.

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
