# Peer review of "Lights and Shadows in Immuno-Oncology Drug Development"

_cancers, 2021, doi:10.3390/cancers13040691_

Round 1

Reviewer 1 Report

In this Perspective, Dr. Sirvén and colleagues present a vision for challenges and opportunities in immune-oncology drug development. They also highlight the importance of developing new and robust predictive biomarkers for the process of patient population selection, drug design and efficacy evaluation to lead approvals. The manuscript is presented very well. The reviewer only has minor comments to the authors.

Minor Comment: The main text is about the “challenges and opportunities”, while the title is “Lights and shadows”. Please be aware.

Typos:

  1. Line 79, the use of-a [sic] relatively reduced number
  2. Lines 123 and 126: iRECIST: Immune [sic] Evaluation Criteria in Solid Tumours. “Response” is missing.
  3. Line 167, response in in [sic] different tumours such
  4. Line 172, and immune infiltrating cells in on [sic] treatment-biopsies
  5. Line 233, companion diagnostic assy [sic] for the administration
  6. Line 230, biomarkers of efficacy of IO drugs to identify patients who benefit from thoese [sic] agents.
  7. Line 254, Multi-plex and mulit [sic]-omics based
  8. Line 263, These signatures are yet to be validated in trials [sic] (“.” is missing)
  9. Line 290, are encouraged in trial desings [sic]
  10. Line 345, incoporating [sic] with multiplex

Author Response

Dear Dr Lila Zhao and collegues,

4th February 2021

Thanks so much for taking the time to review our manuscript ‘Drugs and shadows in inmmuno-oncology drug development’ and for considering it for publication. We have considered the reviewer comments and have revised the manuscript accordingly. Here is the point by point response  to the comments provided by the reviewer detailing all changes that we have made to the original manuscript. Please do let me know if I can improve more.

Reviewer comments:

Reviewer 1:

In this Perspective, Dr. Sirvén and colleagues present a vision for challenges and opportunities in immune-oncology drug development. They also highlight the importance of developing new and robust predictive biomarkers for the process of patient population selection, drug design and efficacy evaluation to lead approvals. The manuscript is presented very well. The reviewer only has minor comments to the authors.

Response: Thank you for your comments and for having noticed the typos and have corrected all the listed below. We also did a comprehensive proof-read for the entire paper again to look for additional mistakes.  Therefore, we had not provided a point to point answer.

Minor comment: The main text is about the “challenges and opportunities”, while the title is “Lights and shadows”.  

  1. Line 79, the use of-a [sic] relatively reduced number. I have deleted the - .
  2. Lines 123 and 126: iRECIST: Immune [sic] Evaluation Criteria in Solid Tumours. “Response” is missing. Response: I have added the word ‘Response’ to ‘Immune Response Evaluation Criteria in Solid tumours’ in both lines.
  3. Line 167, response in in [sic] different tumours such. Response: I have deleted one of the repeated ‘in’.
  4. Line 172, and immune infiltrating cells in on [sic] treatment-biopsies. Response: I have deleted ‘in’ and modified slightly the sentence for more clarity.
  5. Line 233, companion diagnostic assy [sic] for the administration. Response: I have corrected the word spelling mistake to ‘assay’.
  6. Line 230, biomarkers of efficacy of IO drugs to identify patients who benefit from thoese [sic] agents. Response: I have corrected the word ‘those’.
  7. Line 254, Multi-plex and mulit [sic]-omics based. Response: I have corrected it to Multi-plex and multi-omics.
  8. Line 263, These signatures are yet to be validated in trials [sic] (“.” is missing). Response: I have added the final “.”. I have also added clinical to trials, as the sentence sounds clearer.
  9. Line 290, are encouraged in trial desings [sic]. Response: I have corrected ‘designs.’
  10. Line 345, incoporating [sic] with multiplex: Response, I have changed it to ‘multiple’.

Reviewer 2 Report

Nice perspective in the manuscript titled, "Lights and shadows in immuno-oncology drug development", by Bergamino et al., I do not find any specific points that needs attention in the submitted draft. I recommend the manuscript to be accepted for publication. I suggest a careful checking for typos in the draft. (line 188).

Author Response

Reviewer 2:  

Nice perspective in the manuscript titled, "Lights and shadows in immuno-oncology drug development", by Bergamino et al., I do not find any specific points that needs attention in the submitted draft. I recommend the manuscript to be accepted for publication. I suggest a careful checking for typos in the draft. (line 188).

Response: Thank you very much for your comments. We have carefully checked for all the typos in the draft. We have also corrected the word to read “endopoint” in line 188.

Reviewer 3 Report

Immunotherapies have been a major breakthrough in oncology and is being extensively used to treat many cancers. Definitely they have greatly improved tumor outcomes and patient survival, however they is several gaps in how the agents efficacy are evaluated, their longterm effect are studied. In this article authors reviewing these issue and also suggest few remedies.  Overall this article is very well written, help in designing future studies, and will be interesting for readers.    

Author Response

Reviewer 3:

Immunotherapies have been a major breakthrough in oncology and is being extensively used to treat many cancers. Definitely they have greatly improved tumor outcomes and patient survival, however they is several gaps in how the agents efficacy are evaluated, their longterm effect are studied. In this article authors reviewing these issue and also suggest few remedies.  Overall, this article is very well written, help in designing future studies, and will be interesting for readers.    

Response: Thank you for your comments. We are also convinced about the need of improvement in immune-oncology trials designs. This perspective may be helpful for further studies assessing immunotherapeutic treatments. We have also carefully checked for typos and English language in the draft. 

Reviewer 4 Report

"Lights and shadows in immunology-oncology drug development", presents an interesting perspective on the development of immunology-oncolgy drugs and potential efficiency improvements in clinical trial design.

The manuscript will be of interest to a wide readership, is generally well presented, and makes sound recommendations for the field.

Before publication, I would recommend a thorough review of the manuscript for typos/grammar errors, and some revision for brevity and clarity. I have no major concerns regarding this manuscript, but provide a list below of minor errors that need addressing before publication. Note that the following list is not exhaustive, and does not preclude thorough checking by the authors to ensure all errors are identified.

  1. Abbreviation definition and consistency. The authors must ensure that abbreviations are defined at first use. Once defined, ensure that the abbreviations are used throughout, e.g., IO is not defined in the summary, BC & NSCLC are defined but not always used. Ensure that abbreviations are consistent, e.g., TILs or TILS, PDL1 or PD-L1.
  2. line 65: consider revising this sentence for brevity, e.g., Furthermore, differences between the intrinsic biology of IO and conventional therapies, complicate comparisons during clinical trials.
  3. Line 79: 'use of-a relatively reduced' - 'use of a reduced'.
  4. Line 79-82: Revise sentence for clarity.
  5. Line 80:'succeed' - 'success'. 
  6. Line 82-85: Revise sentence for brevity and clarity.
  7. Line 90-91: 'would have derived in positive results' - 'would have led to positive results'.
  8. Line 94: 'receptors' - 'receptor'.
  9. Line 99: Delete 'in some cancers'.
  10. Line 141: 'stability' - 'stable'.
  11. Line 159: Spelling of Immunotherapy.
  12. Line 160-162: Sentence needs revision.
  13. Line 166: Missing word.
  14. Line 167: Duplicated word.
  15. Line 171-176: Sentence needs revision.
  16. Line 188: Spelling of endpoint.
  17. Line 191: Use of apostrophe on IO drugs is incorrect. Also note that the authors switch between IO drug or IO agent. Choose one term. In this case the sentence would read better as 'Surrogate endpoints for the assessment of IO drug efficacy, have primarily been adopted for cytotoxic and molecular targeted drugs...'
  18. Line 193: 'question remains' - 'questions remain'.
  19. Line 197 (plus others): Spelling of meta analysis.
  20. Line 209-219: This section contains multiple errors and should be revised/checked.
  21. Line 222: 'reduced' - 'reduced'.
  22. Line 236-239: Revise sentence.
  23. Line 248-249: Revise sentence.
  24. Line 270-272: Revise sentence.
  25. Line 294-295: Revise sentence.
  26. Line 297-299: Revise sentence.
  27. Line 305-306: Revise sentence.
  28. The conclusions could be revised to make them stronger. Most of this section is quite generic and the sentence structure is poor. My suggestion would be to include specific recommendations to the field.

Author Response

Reviewer 4

Lights and shadows in immunology-oncology drug development", presents an interesting perspective on the development of immunology-oncolgy drugs and potential efficiency improvements in clinical trial design.

The manuscript will be of interest to a wide readership, is generally well presented, and makes sound recommendations for the field.

Before publication, I would recommend a thorough review of the manuscript for typos/grammar errors, and some revision for brevity and clarity. I have no major concerns regarding this manuscript, but provide a list below of minor errors that need addressing before publication. Note that the following list is not exhaustive, and does not preclude thorough checking by the authors to ensure all errors are identified.

Response: Thanks for these great suggestions. We have followed your advice and reviewed all typos and grammar errors. We have also reviewed and modified the recommended sentences and paragraphs for more brevity and clarity. You can find them in the modified version of the paper following your comments. We have also amended the text in conclusion to make it less generic and more focused on our suggestions for improvement.

  1. Abbreviation definition and consistency. The authors must ensure that abbreviations are defined at first use. Once defined, ensure that the abbreviations are used throughout, e.g., IO is not defined in the summary, BC & NSCLC are defined but not always used. Ensure that abbreviations are consistent, e.g., TILs or TILS, PDL1 or PD-L1. Response: Thank you for that. We have reviewed all abbreviations and ensure we are consistent across the whole text.
  2. line 65: consider revising this sentence for brevity, e.g., Furthermore, differences between the intrinsic biology of IO and conventional therapies, complicate comparisons during clinical trials. Response: Thank you for this comment. We have re-structured this sentence for more clarity as follows: ‘Furthermore, the differences in the intrinsic biology between IO agents and conventional therapies further complicate their comparisons’.
  3. Line 79: 'use of-a relatively reduced' - 'use of a reduced'. Response: Thank you, we have modified it.
  4. Line 79-82: Revise sentence for clarity. Response: We have modified this sentence for more clarity as following. ‘Overall, the main problem in the development of IO agents in this type of cancer has been the lack of biomarker-guided patient selection for trials and the reliance on a reduced number of ‘classic’ biomarkers such as PD-L1. In particular, PD-L1 remains at least insufficient to fully explain the therapeutic success and durable clinical benefit seen in some patients with PD-L1 non-expressing tumours, especially when treated with other IO treatments beyond checkpoint inhibitors.’
  5. Line 80:'succeed' - 'success' Response: we have changed it to ‘success’.
  6. Line 82-85: Revise sentence for brevity and clarity. Response: Thank you for this suggestion. We have modified this sentence for more clarity as follows: ‘However, new genomic alterations such as those in DNA damage response or specific mutated gene pathways have shown promising results as immunomarkers in some translational studies and their validation in clinical trials should be encouraged’.
  7. Line 90-91: 'would have derived in positive results' - 'would have led to positive results'. Response: Thank you for this comment, we have modified that expression to your suggestion, as it is a clearer statement.
  8. Line 94: 'receptors' - 'receptor'. Response: We have changed it to ‘receptor’.
  9. Line 99: Delete 'in some cancers'. Response: We have deleted that last part of the sentence.
  10. Line 141: 'stability' - 'stable'. Response: We have changed stability to ‘stable’.
  11. Line 159: Spelling of Immunotherapy. Response: We have corrected that typo.
  12. Line 160-162: Sentence needs revision. Response: We have modified the paragraph for more clarity to: ‘Although the American Society of Clinical Oncology - Society for Immunotherapy of Cancer (ASCO-SITC) has recently recommended to report responses according to both the conventional RECIST and iRECIST criterions in parallel, this approach is still suboptimal as their evaluation is done independently. Harmonising and integrating both measurement methods into an adequate tool rather than just considering both criterions separately is urgently warranted for more precise measurements of actual responses.’.
  13. Line 166: Missing word. Response: We have modified the sentence for more clarity as follows: ‘Some biomarkers assessed in peripheral blood such as…’
  14. Line 167: Duplicated word. Corrected: Response: We have continued the previous sentence for more clarity, as follows: ‘…such as interleukin 8 (IL-8), tumour circulating DNA (cDNA) or CD8+ memory effector cytotoxic T cells have recently shown to assist tracking immune response in different tumours like NSCLC and melanoma.’
  15. Line 171-176: Sentence needs revision. Response: We have modified the whole paragraph as it was difficult to follow: ‘Given the strong correlation found between on-treatment changes on those biomarkers with the subsequent clinical response, biological assays exploring changes on their expression could lead to promising results. Based on that, decisions could be made on the basis of patient-specific immunological tracking biomarkers. Moreover, the pathological assessment of lymphocytes and immune infiltrating cells in tumour biopsies during treatment could also be considered as a useful monitoring tool of clinical response in clinical trials and help to gain deeper understanding of the changes in tumour characteristics under immunotherapy. However, the application of on-treatment biopsies in clinical practise remains unclear and should be further investigated and refined as it could be considered invasive [38]’.
  16. Line 188: Spelling of endpoint. Response: This typo has been corrected.
  17. Line 191: Use of apostrophe on IO drugs is incorrect. Also note that the authors switch between IO drug or IO agent. Choose one term. In this case the sentence would read better as 'Surrogate endpoints for the assessment of IO drug efficacy, have primarily been adopted for cytotoxic and molecular targeted drugs...' Response: Thank you for this suggestion. Following your comment, we have modified the sentence to: 'Surrogate endpoints for the assessment of IO agents’ efficacy, have primarily been adopted for cytotoxic and molecular targeted drugs...'. We have also reviewed and modified the use of drugs and agents after IO for more consistency.
  18. Line 193: 'question remains' - 'questions remain'. Response: This mistake has been changed.
  19. Line 197 (plus others): Spelling of meta-analysis. Response: These typose have been modified all the times.
  20. Line 209-219: This section contains multiple errors and should be revised/checked. Response: Thank you for noticing this error. We have completely modified that paragraph for a better understanding: ‘Recent studies have also shown that endpoints taking into account a component of duration of response such as milestone survival or durable response rate might capture better the delayed and persistent responses derived from IO and should be further studied [46].  For instance, milestone survival is the survival probability at a given time point, defined a priori as two years and durable response rate can be measured as continuous response, such as complete or partial objective responses, beginning within 12 months of treatment and lasting ≥6 months [47]. The advantages of such type of integrative endpoints are that they take into account particular behaviours seen exclusively in IO and allow the use of pre-defined cut-off time intervals that lead to rapid characterisation of survival probability and inference to long-term survival data’.
  21. Line 222: 'reduced' - 'reduced'. Reponse: This typo has been corrected.
  22. Line 236-239: Revise sentence. Response: Thank you for this comment. We have re-written the whole sentence for better understanding and clarity: ‘Lots of efforts are currently focusing on the reproducibility and standardisation of laboratory protocols for the IHC assessment. However, the evaluation of single IHC biomarkers does not completely explain the heterogeneity of tumours and their use seem to be suboptimal in the ‘genetics and omics era’. In particular multiplex diagnostic assays would be more ideal, especially when testing IO agents beyond immune checkpoint inhibitors’.
  23. Line 248-249: Revise sentence. Response: We have modified the sentence for more clarity to: ‘Thus, the implementation and standardisation of robust bioinformatic methodologies and analytical techniques across laboratories are necessary’.
  24. Line 270-272: Revise sentence. Response: We have modified the whole paragraph to: ‘Immunotherapies have the potential to induce toxicity profiles distinct to those from other cancer treatments. Immune-related adverse events are often underdiagnosed, as patients can remain asymptomatic for long periods of time [61,62]. In addition, toxicity is mostly inflammatory-related and assumptions from cytotoxic or molecular targeted treatments are not appropriate for IO agents. In particular, precise considerations must be taken during their development as they do not fit the dose-response/dose-toxicity relationships seen with cytotoxic therapies’.
  25. Line 294-295: Revise sentence. *
  26. Line 297-299: Revise sentence.* Response: We have modified the whole paragraph to make it more understandable as follows: ‘Aforementioned, defining doses and schemes of IO agents seems relatively more complicated than with cytotoxic drugs. New approaches in trial designs including the optimal dosages that will be carried over later phases of drug development are urgently needed. Finally, the incorporation of additional endpoints especially validated for IO agents in early phases of trials for dose selection to improve efficacy and reduce toxicity are also warranted’.
  27. Line 305-306: Revise sentence. Response: We have modified this sentence to: ‘The anti IDO1 drug epacadostat, which was evaluated in a late-phase trial in melanoma, is an example of such conundrum.’
  28. The conclusions could be revised to make them stronger. Most of this section is quite generic and the sentence structure is poor. My suggestion would be to include specific recommendations to the field.

Response: Thank you very much for this suggestion. We have amended the whole conclusion paragraph for further brevity and clarity and to make a stronger statement.

Thank you again for your suggestions and consideration. 

Sincerely, 

Milana Bergamino Sirvén and Maggie Cheang
